# Aggressiveness and Fumonisins Production of *Fusarium Subglutinans* and *Fusarium Temperatum* on Korean Maize Cultivars

**Setu Bazie Tagele, Sang Woo Kim, Hyun Gu Lee and Youn Su Lee ***

Department of Applied Plant Sciences, Kangwon National University, Chuncheon 24341, Korea;
setubazie@gmail.com (S.B.T.); ksw80@kangwon.ac.kr (S.W.K.); mye1991@kangwon.ac.kr (H.G.L.)
* Correspondence: younslee@kangwon.ac.kr; Tel.: +82-33-250-6417; Fax: +82-33-243-3314

**Abstract:** *Fusarium* root rot and stalk rot are becoming a threat to maize production worldwide. However, there is still limited information about the aggressiveness of *Fusarium subglutinans* Edwards and *Fusarium temperatum* and their relationship with fumonisin production. In this study, for the first time, the reaction of seven Korean maize cultivars to *F. subglutinans* and *F. temperatum* was investigated. The results showed that among the maize cultivars, Hik-chal and Miheung-chal had the highest *Fusarium*-induced root rot and stalk rot severity, while De Hack-chal had the lowest disease severity regardless of the *Fusarium* species. Furthermore, the disease resistant cv. De Hack-chal accumulated low levels of fumonisins (FUM) in the infected stalk, while cv. Hik-chal and Miheung-chal had the highest level of FUM. It is worth to note that, plants infected with *F. temperatum* had a higher FUM concentration compared to cultivars infected with *F. subglutinans*. The present study shows a significant correlation between stalk rot ratings and FUM levels and it also presents new information about the potential risk of FUM contamination of maize stalk with *F. subglutinans* and *F. temperatum* in South Korea. In addition, enzyme activities like polyphenol oxidase (PPO), peroxidase (POD), and the amount of total phenol content (TPC) were studied in selected susceptible cultivar Miheung-chal and resistant cultivar De Hack-chal. The activity of PPO, POD and concentration of TPC were generally higher in the roots of the resistant cultivar than the susceptible cultivar. Moreover, following inoculation of either *F. subglutinans* or *F. temperatum*, there was a significant increase in PPO and POD activity in the roots of both cultivars. Hence, the information provided in this study could be helpful to better understand the mechanisms of resistance response to infection of *Fusarium* root rot pathogens.

**Keywords:** biochemical changes; fumonisins; *Fusarium subglutinans*; *Fusarium temperatum*; root rot; stalk rot; *Zea mays*

## 1. Introduction

*Fusarium* species have received high attention due to their economic impacts and their health risk associated with mycotoxin contamination [1]. Maize root rot and stalk rot are among the most economically important diseases of maize worldwide [2,3]. Maize root rot and stalk rot have been reported to be caused by a variety of plant pathogens including *Colletotrichum graminicola* (Cesati) Wilson, *Fusarium graminearum* (Schweinitz) Petch, *Fusarium moniliforme* (Sawada) Wollenweber, *Fusarium verticillioides* Wineland, and *Pythium graminicola* Subramanian [3–5].

Two morphologically similar and phylogenetically closely related *Fusarium* species viz., *Fusarium subglutinans*, and *Fusarium temperatum* [6] have been reported to cause seed rot, root rot, seedling blight, stalk rot, and ear rot of maize worldwide [7–9]. The two sister *Fusarium* species were also reported to be toxigenic pathogens of maize [10–12]. *Fumonisins* (sum of B1, B2, B3) are well known mycotoxins

posing the most hazardous threats to human and animal health [13]. However, the toxicogenic potential of the two above-mentioned *Fusarium* species from South Korea has not been reported. Furthermore, the aggressiveness of the two *Fusarium* species to cause root rot on Korean maize cultivars has not yet been assessed.

The biochemical changes in pathogen-inoculated plants have been previously documented [14,15] and showed the relationship between the biochemical changes and host resistance have been documented [15]. PPO, POD, and TPC are among the most commonly reported defense responses during the interaction between a host plant and a pathogen and they have been reported as a marker for resistance to pathogen infection [16–18]. PPO is a nuclear encoded enzyme that oxidation of phenolic substances to form more toxic substances and some of them function in cross-linking reactions [17,19]. The role of peroxidases activity has been reported in an array of defense-related processes, mainly through direct toxic effect and lignifications [20,21]. Phenolic compounds have direct antimicrobial activity and form barriers through cell wall lignifications to prevent pathogen encroachment [18,22]. To our knowledge, the biochemical changes in the roots of maize plants inoculated with two *Fusarium* species have not yet been reported. Therefore, this study aimed to (i) evaluate the aggressiveness of *F. subglutinans* and *F. temperatum* on maize cultivars currently being grown in South Korea using artificial soil and stalk inoculation (ii) assess capability of Korean isolates of *F. subglutinans* and *F. temperatum* to produce fumonisins (sum of B1, B2, B3) in maize stalk following artificial stalk infection and (iii) to characterize biochemical responses of Korean maize cultivars to root infection caused by *F. subglutinans* and *F. temperatum*.

## 2. Materials and Methods

The experiments of this study were performed under greenhouse conditions at $32 \pm 3$ °C, 65–75% relative humidity, 16 h photoperiod at Kangwon National University, South Korea. Maize seedlings were watered when necessary. The experiment was arranged in a factorial randomized complete block design (cultivars, *Fusarium* species) and the experiment was replicated thrice with 30 plants per replication.

### 2.1. Fungal Strains and Plant Materials

Two *Fusarium* species—namely *F. subglutinans* and *F. temperatum* obtained from plant pathology laboratory, Kangwon National University, South Korea—were used. The above-mentioned *Fusarium* species were found to be the most dominant fungal species isolated from stalk rot of maize in Gangwon province, South Korea [23]. Seven Korean maize cultivars were used in this study. The maize cultivars were: Hik-chal, Miheung-chal, Mibeak-2ho, Cho-dang, Chahong-chal, Speed De Hack-chal, and De Hack-chal.

### 2.2. Inoculum Preparation and Plant Inoculation

In this study, the aggressiveness of *F. subglutinans* and *F. temperatum* on Korean maize cultivars was investigated. The aggressiveness of the pathogens to cause stalk rot was carried out following Shin et al. [23]. Briefly, seven-day-old PDA plate surfaces of *F. subglutinans* and *F. temperatum* were flooded with sterile distilled water and were scrapped. Fragments of mycelia were filter removed through double layers of miracloth. The conidia were washed and adjusted to $4 \times 10^6$ conidia$^{-1}$ mL. One milliliter of the conidial suspension was injected into maize stalks of the four-week-old plant using a sterile needle.

The aggressiveness of the test pathogens to cause root rot on selected maize cultivars was carried out following a soil inoculation method adapted from Lanza et al. [7]. In brief, seeds of each cultivar were first surface-sterilized with ethanol for 2 min followed by NaClO solution (0.75% Cl) for 10 min. The seeds were then washed six times with sterile distilled water [24]. In order to disinfect the naturally existing fungi, the seeds were treated with warm distilled water (60 °C) for 5 min [25]. The seeds were then soaked in a spore suspension ($10^7$ spores$^{-1}$ mL) and were incubated in a rotary shaker at $25 \pm 2$ °C for 48 h at 150 rpm. Seeds kept in sterile distilled water were served as non-inoculated control. Pathogen inoculated

and non-inoculated seeds were sown in a plastic pot in a growth chamber at $28 \pm 2$ °C and 70% humidity. After three weeks of pathogen inoculation, data for root rot severity were recorded.

## 2.3. Disease Assessment

The aggressiveness of *F. subglutinans* and *F. temperatum* to cause root rot on the tested Korean maize cultivars was assessed three weeks after inoculation. Root rot was recorded based on a 0–4 scale, where 0, healthy root; 1, less than 25% of the root spoiled due to rotting; 2, 25–50% of the root spoiled, evident from dropping of the leaves during daytime; 3, up to 75% of the root damaged, evident from starting of wilt and drying of leaves from bottom to top; 4, complete rotting of the root, completely wilted, dead, and dry plants [26].

Stalk rot was assessed by splitting the stalks of four internodes longitudinally four weeks after inoculation. Severity of stalk rot was determined based on 1 to 5 stalk rot severity scale, where 1 = 0 to 25% rot, 2 = 26 to 50% rot, 3 = 51 to 75% rot, 4 = 76 to 100% rot, and 5 = 100% rot with infection extending into an adjacent internode [27].

For analysis of root rot and stalk rot severity, the scale was converted into percentage severity index (PSI) [28].

$$\text{PSI} = \frac{\sum \text{ of all numerical ratings } \times 100}{\text{Total number of observations } \times \text{ maximum score on scale}} \tag{1}$$

## 2.4. Genotype Classification

The modified rank-sum method was employed to classify the tested Korean maize cultivars into different resistance categories according to the cultivars mean percent severity index (PSI) [29]. For this, PSI of root rot and stalk rot caused by *F. subglutinans* and *F. temperatum* for each maize cultivar was used. In order to calculate the rank-sum, the cultivars were assigned ranks in ascending order based on the mean PSI value obtained from RANK procedure of the SAS program [30]. The ranks (Xn) of each cultivar from root rot and stalk rot caused by *F. subglutinans* and *F. temperatum* were summed and then compared to the grand mean of the rank-sums calculated across all cultivars (Gn). The deviation of each cultivar from the grand mean was computed as ((Xn − Gn)/standard deviation) × 2. Cultivars having a deviation of positive values were categorized susceptible and cultivars with negative deviations were categorized resistant. In brief, cultivars having deviation value of 1, 2, and 3 were classified as moderately susceptible, susceptible, and highly susceptible, respectively, while cultivars with −1, −2, and −3 values were classified as moderately resistant, resistant, and highly resistant, respectively [31].

## 2.5. Determination of Fumonisins (FUM) Using ELISA

Maize stalk samples (in triplicates) were ground to a fine powder with a mill. The concentration of fumonisins (FUM) produced by *F. subglutinans* and *F. temperatum* was quantified using ELISA kits (RIDASCREEN FAST fumonisins (sum of B1, B2, B3); R-Biopharm AG, Darmstadt, Germany). For analysis, 5 g sample of each was extracted with 25 ml of 70% methanol and the mixture was vigorously shaken for 1 hr on a shaker. The extract was filtered through Whatman No. 1 filter and the filtrate was diluted (1:5 and 1:10) with 70% methanol. The diluted extract (50 µl) was used for FUM analysis following the manufacturer's instructions.

## 2.6. Biochemical Changes during Interaction of Maize Root and the Test Pathogens

### 2.6.1. Extraction

After three weeks of pathogen inoculation, fresh root samples were collected separately from inoculated and non-inoculated plants to determine the activities of polyphenol oxidase, peroxidase, and total phenol content due to pathogen infection. The samples were washed with sterile distilled water to remove soil. The samples were immediately frozen in liquid nitrogen and were kept at –80°C until used.

### 2.6.2. Polyphenol Oxidase (PPO) Activity

The PPO activity was determined following the method described by [32]. Briefly, extracts were prepared by homogenizing one-gram sample in 3 ml ice-cold phosphate buffer (0.1 M, pH =6.5) using mortar and pestle. The homogenate was centrifuged at 15,000 rpm for 15 minutes at 4 °C. The PPO assay was comprised of a mixture 2 ml crude extract, 3 ml of 0.1M phosphate buffer (pH = 6.5), and 1.0 ml catechol (0.01 M). The mixture was incubated at $28 \pm 2$ °C for 5 min. Following the beginning of the reaction, the developed yellow color was read using spectrophotometer against a substrate blank at a wavelength of 495 nm at one-minute interval. The experiment was replicated three times. The activity of PPO was determined based on the change in absorbance and was presented as enzyme activity per mg fresh weight per min [33].

### 2.6.3. Peroxidase (POD) Activity

The activity of POD was assayed following the method described by Urbanek et al. [34]. In brief, extracts were prepared by homogenizing one-gram samples in 3 ml ice-cold phosphate buffer (0.1 M, pH = 6.5) using mortar and pestle. The homogenate was centrifuged at 15,000 rpm for 15 minutes at 4 °C. The resulting supernatant was used as an enzyme source for POD analysis. The reaction mixture contained 2.3 ml 50 mM potassium phosphate buffer (6.8), 0.1 ml crude enzyme extract, 0.5 ml guaiacol, and 0.1 ml $H_2O_2$ (5%). The mixture was incubated for 10 min at 30 °C and 0.5 ml $H_2SO_4$ (5% v/v) was added. The activity was determined by measuring the absorbance at 480 nm in 15 seconds interval and the POD activity was presented as expressed in terms of the enzyme activity per mg fresh weight per min. The experiment was replicated three times.

### 2.6.4. Total Phenol Content (TPC)

Total phenol content (TPC) found in the root samples of pathogen-inoculated and non-inoculated maize plant was determined using Folin–Ciocalteau phenol reagent [35]. Root samples (1.0 g) were ground in liquid $N_2$ using a mortar and pestle. The samples were then mixed in 10 ml of 80% methanol and were extracted by shaking at room 70 °C for 15 min. The methanol extract was centrifuged at $4000 \times$ g for 5 min at 42 °C. One ml of the extract was mixed with 5 ml of distilled water and 0.5 ml Folin–Ciocalteau phenol reagent (1:1 diluted) for 3 minutes, followed by addition of 1 ml of 20% sodium carbonate (*w/v*). The mixture was properly stirred and kept at room temperature in dark for 60 min. Subsequently, the intensity of the developed blue color was measured using a spectrophotometer at 720 nm. The experiment was replicated three times. The TPC was calculated from the standard curve of catechol and was expressed as mg of catechol equivalents per gram of root fresh weight.

### 2.7. Statistical Analysis

A simple linear correlation was calculated to determine the relationship between visual stalk rot disease severity and FUM concentrations using the CORR procedure in SAS software version 9.2 (SAS Institute, Cary, NC, USA) [30]. Data on *Fusarium* rot severity and biochemical tests were analyzed using the PROC GLM procedure of SAS software version 9.2 (SAS Institute, Cary, NC, USA) [30]. Means of the simple main effect were separated using post hoc Tukey honestly significant difference (HSD) test. All experiments in the present study were replicated at least three times and the results were expressed as mean ± standard error.

## 3. Results

### 3.1. Aggressiveness of the Two Fusarium Species to Cause Stalk Rot

In the stalk inoculation experiment, stalks of the tested Korean maize cultivars were artificial inoculated with *F. subglutinans* and *F. temperatum*. Subsequently, typical stalk rot symptoms and substantial amount of stalk rot severity were recorded. The results of the study indicated that there was

no significant (*p* > 0.05) interaction between cultivar and the *Fusarium* species for stalk rot (Table S1). There was a highly significant difference among Korean maize cultivars for their susceptibility to stalk rot in both pathogens (Table 1). In *F. subglutinans*-inoculated plants, percent severity index (PSI) ranged from 25.3 to 75.1 with a mean value of 55.8 PSI. On the other hand, *F. temperatum*-inoculated plants had a higher mean PSI of 64.3 (ranging from 30.2 to 90.3) (Table 1). Of seven Korean maize cultivars tested, Hikchal and Miheung-chal had significantly (*p* < 0.05) higher levels of PSI for both *Fusarium* species (Table 1, Figure 1). To the contrary, cultivar De Hack-Chal showed significantly (*p* < 0.05) lower PSI (Table 1, Figure 1) in both *Fusarium* species. In addition, the results showed that *F. temperatum* was slightly more pathogenic than *F. subglutinans* in all cultivars tested as it caused comparatively larger necrotic lesions (Table 1, Figure 1).

**Table 1.** Effect of maize cultivar and *Fusarium* species on intensity of *Fusarium* stalk rot.

| Factors | Treatments | Percent Severity Index [1] |
|---|---|---|
| Cultivar | Chahong-chal | 68.33 ± 3.45 [c] |
| | Cho-dang | 50.00 ± 2.58 [d] |
| | De Hack-Chal | 27.75 ± 2.14 [e] |
| | Hikchal | 82.70 ± 4.44 [a] |
| | Mibeak-2ho | 37.75 ± 3.18 [de] |
| | Miheung-chal | 80.00 ± 3.92 [ab] |
| | Speed De Hack-Chal | 74.55 ± 6.6 [ab] |
| *Fusarium* species | *F. subglutinans* | 55.92 ± 4.91 [b] |
| | *F. temperatum* | 64.41 ± 5.45 [a] |

[1] Percent severity index assessed four weeks after stalk inoculation. Mean values having the same letter (s) in each a column are not statistically different (*p* ≤ 0.05).

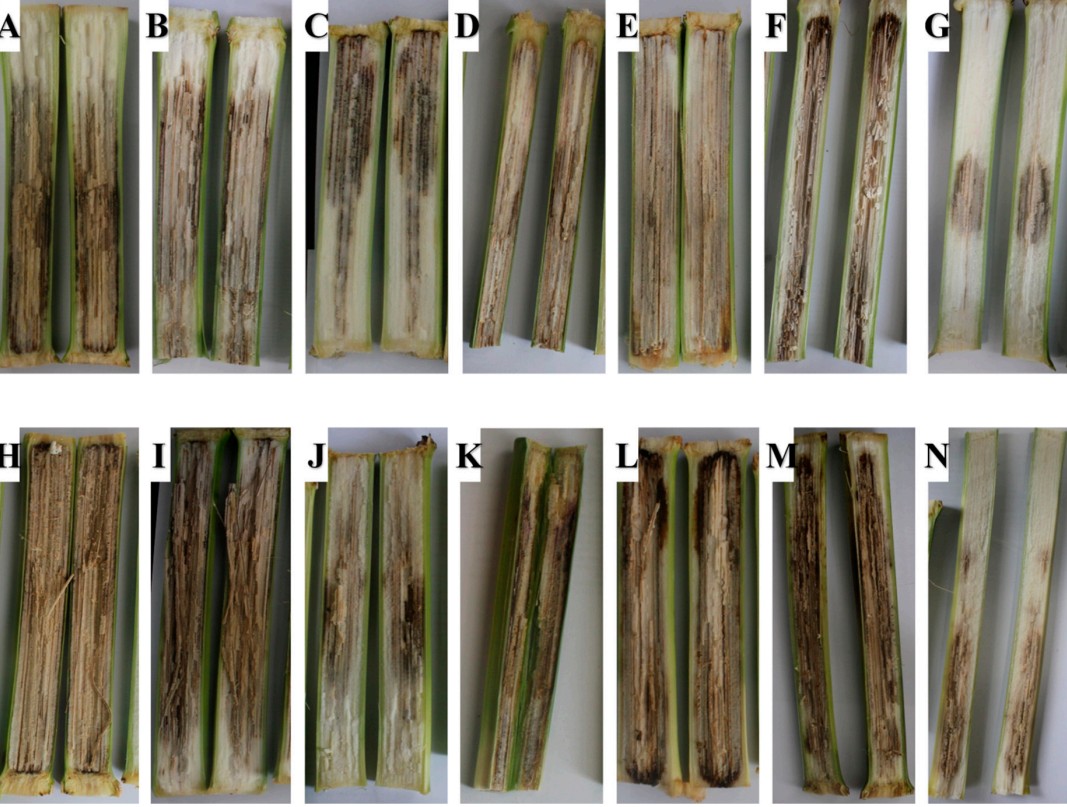

**Figure 1.** Maize stalk rot of cultivar (**A**,**H**) Hik-chal, (**B**,**I**) Miheung-chal, (**C**,**J**) Mibeak-2ho, (**D**,**K**) Cho-dang, (**E**,**L**) Chahong-chal, (**F**,**M**) Speed De Hack-chal, and (**G**,**N**) and De Hack-chal (top, A − G = inoculated with *F. subglutinans*, bottom, H – N = inoculated with *F. temperatum*).

### 3.2. Fumonisin (FUM) Concentration in Maize Stalks

The concentration of FUM produced on maize stalk by the two *Fusarium* species was analyzed on seven Korean maize cultivars using a sandwich enzyme-linked immunosorbent assay (ELISA) test kit (RIDASCREEN® FAST fumonisins (sum of B1, B2, B3); R-Biopharm AG, Darmstadt, Germany). The results of the study revealed that there was no significant interaction between cultivar and *Fusarium* species for FUM concentration (Table S1). In addition, plants infected with *F. temperatum* had no significant mean FUM concentration (1558.8 $\mu$g g$^{-1}$) compared to *F. subglutinans* (1518.8 $\mu$g g$^{-1}$) in the stalk of the tested maize cultivars. Among the cultivars tested, cv. De Hack Chal accumulated low level of FUM in either *F. subglutinans* or *F. temperatum* inoculations. On the other hand, cv. Hikchal, Miheung-chal, Chahong-chal, and Speed De Hack Chal had substantially higher concentration of FUM, regardless of the *Fusarium* species (Table 2). Furthermore, this study indicated that there was positive correlation between stalk rot severity and FUM concentrations on each cultivar for *F. subglutinans* ($r$ = 0.8269) and *F. temperatum* ($r$ = 0.8243).

**Table 2.** Effect of maize cultivar and *Fusarium* species on fumonisins concentrations (in $\mu$g g$^{-1}$) produced in maize stalks.

| Factors | Treatments | Fumonisins Concentrations ($\mu$g g$^{-1}$) [A] |
|---|---|---|
| Cultivar | Chahong-chal | 1795.15 ± 24.82 [a] |
| | Cho-dang | 1404.12 ± 21.31 [b] |
| | De Hack-Chal | 887.70 ± 25.49 [d] |
| | Hikchal | 1835.47 ± 16.53 [a] |
| | Mibeak-2ho | 1227.85 ± 26.39 [c] |
| | Miheung-chal | 1815.65 ± 21.57 [a] |
| | Speed De Hack-Chal | 1805.23 ± 34.75 [a] |
| *Fusarium* species | *F. subglutinans* | 1518.83 ± 151.90 [a] |
| | *F. temperatum* | 1558.79 ± 139.79 [a] |

[A] Fumonisins concentrations assessed four weeks after stalk inoculation. Mean values having the same letter (s) in each a column are not statistically different ($p \leq 0.05$).

### 3.3. Aggressiveness of the Two Fusarium Species to Cause Root Rot

In the soil inoculation method, the aggressiveness of *F. subglutinans* and *F. temperatum* to cause root rot on the seedlings of seven Korean maize cultivars was assessed three weeks after soil inoculation. Soil inoculation either with *F. subglutinans* or *F. temperatum* resulted in typical discolored symptoms on primary and secondary roots of the tested cultivars (Figure 2). The results of the study indicated that there was no significant ($p$ >0.05) interaction between cultivar and the *Fusarium* species for root rot severity (Table S1). The root rot severity data showed that there was a highly significant difference between the tested Korean maize cultivars (Table 3). Among the tested cultivars, cv. De Hack-chal was found to be comparatively resistant to the two *Fusarium* species as indicated by low infection level (Table 3 and Figure 2). In contrast, cv. Hik-chal and Miheung-chal had the highest *Fusarium*-induced root rot severity as indicated by severely discolored roots, regardless of the *Fusarium* species (Table 3 and Figure 2). The remaining cultivars viz., Mibeak-2ho, Cho-dang, Chahong-chal, and Speed De Hack-chal showed comparatively mild infection with the tested *Fusarium* root rot pathogens (Table 3). Furthermore, *F. temperatum* was more aggressive than *F. subglutinans* (Table 3).

**Table 3.** Effect of maize cultivar and *Fusarium* species on intensity of *Fusarium* root rot.

| Factors | Treatments | Percent Severity Index [1] |
|---|---|---|
| Cultivar | Chahong-chal | 11.35 ± 1.33 [b] |
| | Cho-dang | 9.65 ± 0.95 [b] |
| | De Hack-Chal | 3.10 ± 0.68 [c] |
| | Hikchal | 21.00 ± 1.84 [a] |
| | Mibeak-2ho | 8.00 ± 1.15b [c] |
| | Miheung-chal | 18.65 ± 0.67 [a] |
| Fusarium species | Speed De Hack-Chal | 12.65 ± 1.23 [b] |
| | F. subglutinans | 10.12 ± 2.34 [b] |
| | F. temperatum | 14.00 ± 2.65 [a] |

[1] Percent severity index assessed three weeks after soil inoculation. Mean values having the same letter (s) in each a column are not statistically different ($p \leq 0.05$).

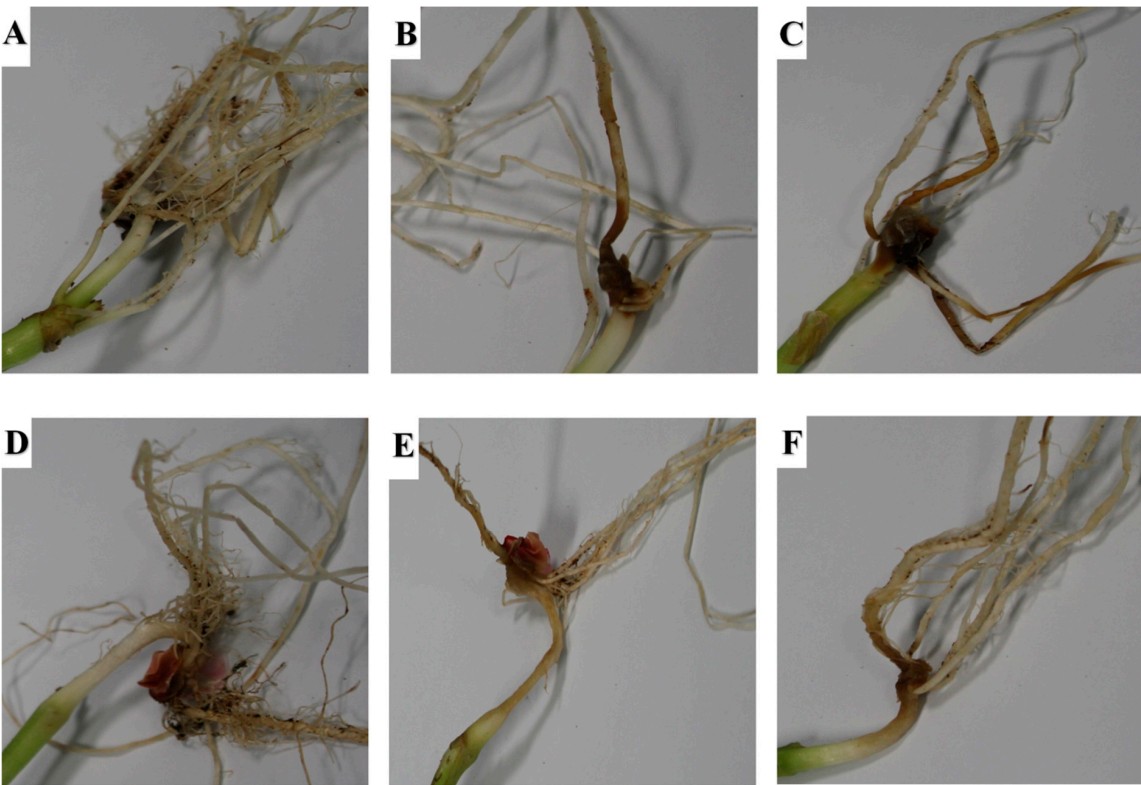

**Figure 2.** Pictorial view of root inoculated with (**A**,**D**) no pathogen, (**B**,**E**) *F. subglutinans*, and (**C**,**F**) *F. temperatum*) (top, **A**–**C** = Miheung-chal, bottom, **D**–**F** = De Hack-Chal).

### 3.4. Biochemical Changes During Interaction of Maize Root and the Two Fusarium species

In this study, the biochemical changes of the selected resistant cultivar De Hack-Chal and susceptible cultivar Miheung-chal Korean maize cultivars due to infection caused by *F. subglutinans* and *F. temperatum* were studied (Figure 3). In non-inoculated plants, the resistance cultivar De Hack-chal had a higher PPO activity than the susceptible cultivar Miheung-chal (Figure 3). More importantly, the PPO activity was increased when plants were inoculated with either of the two *Fusarium* species in both cultivars tested (Figure 3). The resistant cultivar De Hack-chal showed the higher concentration of PPO compared to the susceptible cultivar Miheung-chal due to pathogen inoculation. The activity varied from 6.1 (non-inoculated plants) to 12.1 (*F. temperatum*-inoculated plants) in the resistance cultivar De Hack-chal. Similarly, the activity of POD was significantly increased in the pathogen-inoculated plants compared to non-inoculated plants in both cultivars (Figure 3). Maize plants inoculated with *F. temperatum* had higher peroxidase activity compared to *F. subglutinans*

(Figure 3). Furthermore, maximum peroxidase was obtained in *F. temperatum*-inoculated resistant cultivar De Hack-chal followed by *F. temperatum*-inoculated susceptible cultivar Miheung-chal (Figure 3). In addition, the data of total phenol content revealed that resistant cultivar De Hack-chal had significantly higher content than the susceptible cultivar Hik-chal (Figure 3). More importantly, unlike PPO and POD, the TPC was not enhanced in pathogen-inoculated plants than non-inoculated plants in both cultivars.

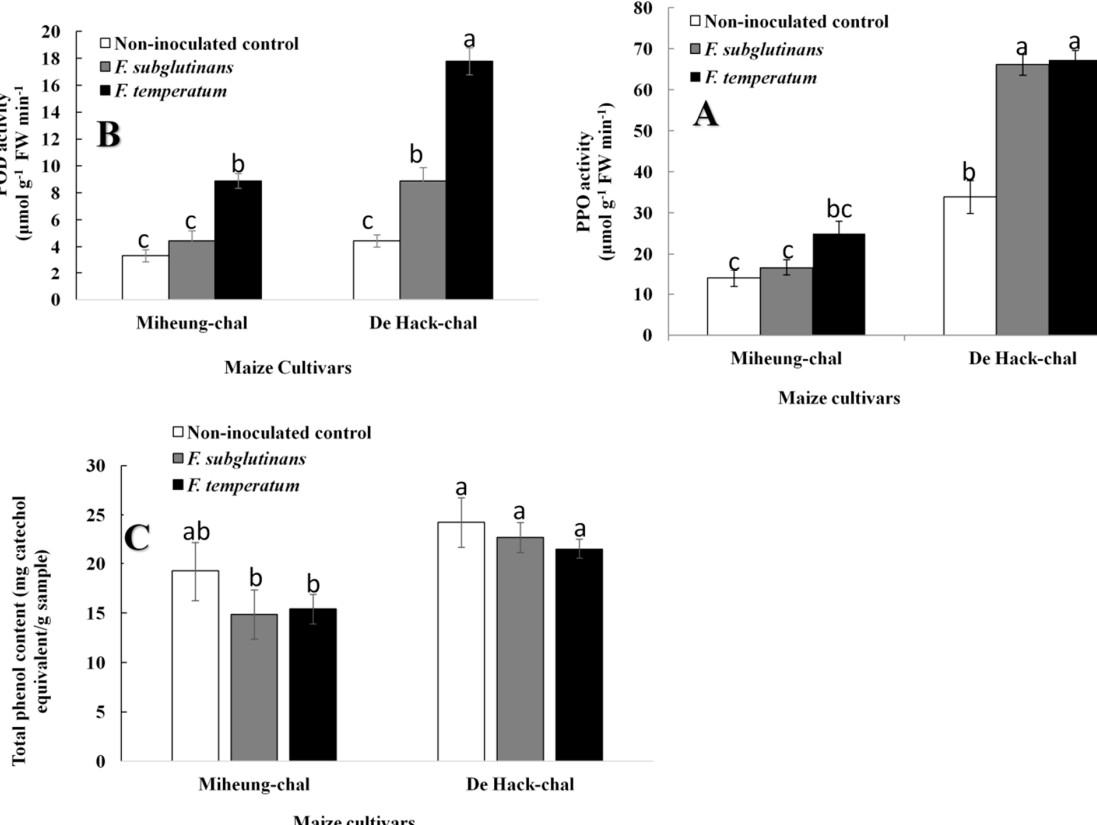

**Figure 3.** Activity of (**A**) polyphenol oxidase, (PPO) (**B**) peroxidase POD and concentration of (**C**) total phenol content (TPC) of inoculated (*F. subglutinans* and *F. temperatum*) and non-inoculated plants of susceptible (Miheung-chal) and resistant (De Hack-chal) cultivars. Mean values having the same letter (s) in each cultivar are not statistically different ($p \leq 0.05$).

## 3.5. Cultivar Ranking

Based on modified rank-sum analysis, cv. De Hack-Chal, Mibeak-2ho and Cho-dang were categorized under highly resistant, resistant and moderately resistant, respectively to root rot and stalk rot caused by either *F. subglutinans* or *F. temperatum*. On the other hand, cv. Hikchal and Miheung-chal were categorized under highly susceptible and susceptible, respectively to root rot and stalk rot caused by the two *Fusarium* species. Cultivar Chahong-chal and Speed De Hack-Chal were found to be moderately susceptible to root rot and stalk rot caused by either *F. subglutinans* or *F. temperatum* (Table 4).

**Table 4.** Rank-sum classification of Korean maize cultivars with different levels of *Fusarium* root rot and stalk rot resistance [1].

| Cultivars | Root Rot | | Stalk Rot | | Cultivar Ranking | | Class [3] |
|---|---|---|---|---|---|---|---|
| | PSI [1] | a [2] | PSI [1] | b [2] | c [2] | d [2] | |
| Hikchal | 21.0 | 7 | 82.5 | 7 | 14 | 2.8 | HS |
| Miheung-chal | 18.7 | 6 | 80.0 | 6 | 12 | 1.9 | S |
| Chahong-chal | 11.3 | 4 | 68.3 | 4 | 8 | 0.0 | MS |
| Speed De Hack-Chal | 12.7 | 5 | 74.3 | 5 | 10 | 0.9 | MS |
| Cho-dang | 9.7 | 3 | 50.0 | 3 | 6 | −0.9 | MR |
| Mibeak-2ho | 8.0 | 2 | 37.5 | 2 | 4 | −1.9 | R |
| De Hack-Chal | 3.1 | 1 | 27.5 | 1 | 2 | −2.8 | HR |
| Grand mean (G) [4] | | | | | 8 | | |
| Standard deviation | | | | | 4.32 | | |

[1] PSI = percent severity index, mean of PSI of *F. subglutinans* and *F. temperatum* data. [2] a = cultivar ranking based on root rot PSI; b = cultivar ranking based on stalk rot PSI; c = rank−sum (a + b) for each cultivar; d = deviation from the grand mean (G) of the rank-sums ((d = (c −G)/standard deviation) × 2). [3] HR=highly resistant, R = resistant, MR = moderately resistant, MS = moderately susceptible, S = susceptible, and HS = highly susceptible. [4] Grand mean of the rank-sums (G).

## 4. Discussion

In this study, the aggressiveness of *F. subglutinans* and *F. temperatum* on the seven Korean maize cultivars was assessed using soil and stalk inoculation methods. Previous studies [4] suggested artificial stalk inoculation to select stalk rot resistant maize genotypes as it caused the required disease pressure to identify the resistant genotypes and it mimics the entrance of pathogen inoculum through mechanical wounds caused by birds, insects, and mechanical damage [36]. Hence, in this study, following artificial stalk inoculation of maize stalks with *F. subglutinans* and *F. temperatum*, cultivar Hik-chal and Miheung-chal were found to be susceptible, while cultivar De Hack-Chal was resistant to the tested pathogens. More importantly, *F. temperatum* was comparatively more pathogenic than *F. subglutinans* in all cultivars tested as it caused comparatively larger necrotic lesions. In agreement with our study, previous studies [10,11] reported that *F. temperatum* has high pathogenic property than *F. subglutinans*.

The result of ELISA test revealed that plants infected with *F. temperatum* had higher mean FUM concentration compared to *F. subglutinans* in all maize cultivars tested. Similarly, the previous studies [10,11] reported a higher toxicological risk of *F. temperatum* than *F. subglutinans*. Furthermore, previous reports [10,37,38] noted the potential of *F. subglutinans* and *F. temperatum* to produce FUM. Among the tested cultivars, cultivar De Hack Chal accumulated comparatively low levels of FUM in the infected stalks, while cultivar Hikchal and Miheung-chal had a substantially higher concentration of FUM in the infected stalks, regardless of the *Fusarium* species (Table 4). Remarkably, the level of FUM in the susceptible Korean maize cultivars was close to the recommended lowest FUM limit for unprocessed corn ($2000 \ \mu g \ g^{-1}$) [39]. Scauflaire and his co-workers [10] reported the potential risk of FUM in the maize stalk produced by the two *Fusarium* species to animal feed. Such results signify that urgent measures should be taken to enhance overall levels of stalk rot resistance in Korean maize cultivars and subsequent prevention of FUM contamination hazard before it become prevalent in the South Korea. Furthermore, the positive correlation between stalk rot severity and FUM concentrations on each cultivar in both *Fusarium* species was observed. Similarly, previous studies [40–42] reported that a reduction in disease symptom development was correlated with lower mycotoxin level. This might be due to the ability of resistant cultivars to degrade the already produced mycotoxins or to reduce mycotoxin production by the pathogen [40]. Determining the level of FUM is expensive and it is unfeasible in routine breeding activity [43]. Similarly, a highly significant correlation between visual rot symptoms and FUM content has been previously reported [44,45]. Furthermore, Butrón et al. [46] reported the Fum contamination of the tested genotypes was similar across environments. Hence, the present study showing the significant correlation between stalk rot

ratings and FUM levels signifies that *Fusarium* stalk rot ratings can be an alternative tool to indirectly select maize cultivars or breeding lines having low FUM. In contrast, the previous report [47] noted that there is an interaction effect of cultivars and environment for resistance to both rot and fumonisin contamination, indicating FUM production is environment dependent [46,47].

Based on soil inoculation method, both *F. temperatum* and *F. subglutinans* resulted in typical discolored symptoms [10,48] on primary and secondary roots of the tested cultivars viz., Miheung-chal and De Hack-Chal. In addition, *F. temperatum* was found to be more aggressive than *F. subglutinans* in causing root rot. Our result is in agreement with the previous report [7,10,11] that high disease severity was recorded in *F. temperatum*-plants than *F. subglutinans*-plants. However, our study reports for the first time about maize root rot in Korean maize cultivars caused by *F. subglutinans* and *F. temperatum*. Based on modified rank-sum analysis, cultivar De Hack-Chal was categorized under highly resistant, while cultivars Hikchal and Miheung-chal were categorized under highly susceptible and susceptible, respectively to *F. subglutinans* and *F. temperatum*. Interestingly, the modified rank-sum method [31] has been reported to have many advantages as it does not require prior knowledge of the genetic information of the test cultivars.

The results of biochemical study revealed that there was a significant increase in PPO and POD activity in the roots of both susceptible Miheung-chal and resistant cultivar De Hack-Chal following inoculation of either *F. temperatum* or *F. subglutinans* compared to non-inoculated plants. Similar results were found in the previous reports [49,50] that PPO and POD activities due to pathogen attack were significantly higher compared to non-inoculated plants. Previous studies [33,51] reported enhanced PPO activity led to high resistance in different plant-pathogen interactions. PPO is an enzyme that catalyzes the oxidation of phenolics to free radicals [50]. The resulting free radicals, in turn, cause an inconvenient environment for the development of a pathogen [36]. An increase in peroxidase due to pathogen challenge has been previously reported [52–54]. Interestingly, an increase in peroxidase activity has been associated with an enhanced defense response [55–57]. POD is one of the vital antioxidant enzymes involved in the production of reactive oxygen species, which directly kills pathogens via its toxic effect and inhibits pathogen spread through lignin production [58,59].

In addition, the TPC in the resistant cultivar De Hack-chal was comparatively higher than the susceptible cultivar Hik-chal. Similarly, several authors [54,60] reported that resistant cultivars were rich in phenol content, while cultivars with low phenolic acid were susceptible to pathogen attack. Phenolic compounds have direct antimicrobial activity and are precursors for the synthesis of structural polymers such as lignin which help to block pathogen spread [61]. It is worth to note that the TPC of plants upon pathogen-inoculation was reduced in both cultivars. This may be due the utilization of phenols in the defense mechanisms including cell wall lignification [62] and fungitoxic quinones [63]. Nevertheless, the reduced phenol content in the pathogen-inoculated plants was balanced by higher activity of peroxidases [62].

## 5. Conclusions

In the present study, the aggressiveness of *F. subglutinans* and *F. temperatum* on Korean maize cultivars was investigated using soil and stalk inoculation methods and the subsequent fumonisins (FUM) concentration of maize stalk was determined. The results revealed that *F. temperatum* was comparatively more pathogenic than *F. subglutinans* in all cultivars tested. Plants infected with *F. temperatum* had higher mean FUM concentration compared to *F. subglutinans* in the tested cultivars. Our study presents new information about maize stalk FUM contamination with *F. subglutinans* and *F. temperatum* in South Korea. Cultivar De Hack-chal was more resistant to root rot and stalk infection, while cv. Hik-chal and Miheung-chal had the highest *Fusarium*-induced root rot and stalk rot severity as indicated by severely discolored roots and larger necrotic lesions. Enzyme activities like PPO and POD were significantly elevated following artificial inoculation of the resistant cultivar De Hack-chal with *F. subglutinans* and *F. temperatum*. Hence, the information provided in this study can be helpful to better understand the mechanisms of resistance response to infection of *Fusarium* root rot pathogens.

**Supplementary Materials:** The following are available online at http://www.mdpi.com/2073-4395/9/2/88/s1, Table S1: Mean square values from two-way analysis of variance for visual severity of root rot, stalk rot, and FUM concentration on Korean maize cultivars (d.f.: degrees of freedom; FUM: Fumonosins).

**Author Contributions:** Conceptualization, S.B.T. and Y.S.L.; Methodology, S.B.T., S.W.K., H.G.L. and Y.S.L.; Software, S.B.T., S.W.K., and Y.S.L.; Validation, S.B.T., S.W.K., H.G.L. and Y.S.L.; Formal Analysis, S.B.T., S.W.K., and Y.S.L.; Investigation, S.B.T., S.W.K., H.G.L. and Y.S.L.; Resources, Y.S.L.; Data Curation, S.B.T. and Y.S.L.; Writing-Original Draft Preparation, S.B.T. and Y.S.L.; Writing-Review & Editing, S.B.T. and Y.S.L.; Visualization, S.B.T. and Y.S.L.; Supervision, S.B.T., S.W.K., H.G.L. and Y.S.L.; Project Administration, H.G.L. and Y.S.L.; Funding Acquisition, Y.S.L.

**Funding:** This research work was financially supported by University Industry Cooperation Foundation of Kangwon National University.

**Acknowledgments:** We are grateful to the University Industry Cooperation Foundation of Kangwon National University for the financial assistance.

**Conflicts of Interest:** The authors declare no conflict of interest.

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
