# Peer review of "Aggressiveness and Fumonisins Production of Fusarium Subglutinans and Fusarium Temperatum on Korean Maize Cultivars"

_agronomy, doi:10.3390/agronomy9020088_

Reviewer 1 Report

The paper reports the the susceptibilty of different Korean maize cultivars to two Fusarium species producing root and stalk rots.
The research design and the methods used are appropriate and adequately described.

I have two concerns about the paper:

The introduction must explain why PPO, POD and phenolic content have been selected as markers to distinghish among suscebtible and resistant genotypes in control and infected plants. It is well known that after Fusarium infection several metabolites are accumulated (see for example the recent review Frontiers in Plant Sciences  8:1774. doi: 10.3389/fpls.2017.01774)

The correlation are not presented in the correct form. I suggest to delete Fig. 2 because presents data already reported in Tables 1 and 2 and to add the R2 values at line 207. 

Author Response

Response to reviewers

Dear Editor and Reviewers,

Thank you very much for your quick response and valuable comments provided. We agree with all comments and we have revised the manuscript accordingly. Please find below a point-by-point response to the reviewers’ comments (in red).

We look forward to hearing from you soon.

Sincerely,

Youn Su Lee, Ph.D., Professor

College of Agriculture and Life Sciences

Kangwon National University, Chuncheon 24341

Republic of Korea (South Korea)

Tel: (+82)-33-250-6417, CP: (+82)-10-4131-6417

[email protected][email protected]

Vice-Chairman, Asian Mycological Association

Korean Representative, International Mycological Association

Past President, Korean Society of Mycology

Past Vice-President, Korean Society of Plant Pathology

We appreciate the reviewer’s comments and the followings are our point-by-point responses.

Response to Reviewer 1 Comments

Point 1: English language and style are fine/minor spell check required

Response 1: we have carefully revised the manuscript. We hope we have improved it.

Point 2: I have two concerns about the paper:

The introduction must explain why PPO, POD and phenolic content have been selected as markers to distinghish among suscebtible and resistant genotypes in control and infected plants. It is well known that after Fusarium infection several metabolites are accumulated (see for example the recent review Frontiers in Plant Sciences 8:1774. doi: 10.3389/fpls.2017.01774)

The correlation are not presented in the correct form. I suggest to delete Fig. 2 because presents data already reported in Tables 1 and 2 and to add the R2 values at line 207.

Response 2: we agree with the reviewer that it is important to have an explanation in the introduction about the necessity of PPO, POD and phenolic content in plant-pathogen interactions. Hence, the following sentence has been added in the revised manuscript (Page 2: lines 58-65).

PPO, POD and TPC are among the most commonly reported defense responses during the interaction between a host plant and a pathogen and they have been reported as a marker for resistance to pathogen infection [16-18]. PPO is a nuclear encoded enzyme that oxidation of phenolic substances to form more toxic substances and some of them function in cross-linking reactions [17, 19]. The role of peroxidases activity has been reported in an array of defense-related processes, mainly through direct toxic effect and lignifications [20, 21]. Phenolic compounds have direct antimicrobial activity and form barriers through cell wall lignifications to prevent pathogen encroachment [18, 22]. ”.

With regard to the correlation graph, we agree that the graph normally talks about the two tables, table 1 and table 2. However, we believe that the graph enable the readers to easily catch-up the relationship about the severity of stalk rot and fumonisin accumulation in the tested Korean maize cultivars. Thus, to satisfy the reviewer concern, we have deleted the graph. Furthermore, we have added the r value (based on reviewer 2 request R2 changed to r) in the paragraph following the reviewer request (Page 6: lines 276-77).

“cultivar for F. subglutinans (r=0.8269) and F. temperatum (r=0.8243)”

Response to Reviewer 2 Comments

Point 1: Moderate English changes required

Response 1: we have carefully revised the manuscript and corrected the grammatical mistakes. We hope we have improved it.

General comments

Point 2: There are still a lot of typos in the manuscript, below I listed some of them. So, please read the manuscript carefully to avoid typos.

Response 2: we really appreciate the reviewer for detail review and we have carefully revised the manuscript accordingly.

Point 3: In addition, you performed the experiments with one isolate from both species. I would be better to include more isolates, to see the differences between isolates.

Response 3: We would like the reviewer know that our base for choosing the two Fusarium species is based on the previous study [23] that the mentioned species have been found the most dominant in the study area. Hence, to avoid confusion, the following sentence has been added in the revised manuscript (Page 2: lines 80-82).

“The above-mentioned Fusarium species were found to be the most dominant fungal species isolated from stalk rot of maize in Gangwon province, South Korea [23]”.

Similar to this question, the reviewer has asked the the origin of the isolates, where and when were they collected as it was indicated in the specific comments L62.

We agree with the reviewer that it is necessary to describe the origin of the fungal species. In addition, it is important to note that our base for choosing the two Fusarium species is based on the previous study [23] that the mentioned species have been found the most dominant in the study area.

Point 4: Your FUM levels are never higher than 2000 µg/g, so even in case the disease pressure is very high the values are never above the limit, so is there a problem? What are the average infection levels expected under field conditions?

Response 4: The level of Fum presented in our study is a mean value fumonisins (sum of B1, B2, B3). Obviously, the level of FUM in the present study by the study Fusarium species can be taken as a high FUM value compared to the previous FUM reports by the same species. For instance, Wang and his co-workers (2013) reported 166.89 μg/g FUM in maize grain sample, with massively more FB2 mycotoxin (2.8 to 108.8fold) than FB1. However, it is important to note that mycotoxin contamination like FUM is dependent on environment, and isolated hosts.

Wang and his co-workers (2013): https://onlinelibrary.wiley.com/doi/pdf/10.1111/jph.12164

Specific comments

 L12 with fumonosins production

Response: corrections have been made (Page 1: line 12). Morover, the word fumonosin was spelt wrongly in the original manuscript but now it has been spelt correctly as fumonisin in the revised manuscript.

L13 to artificially inoculated F. subglutinans and

Response: corrections have been made (Page 1: line 13).

L14 maize cultivars tested

Response: corrections have been made (Page 1: line 14).

L15 stalk rot severity, while

Response: fixed (Page 1: lines 17).

L17 low levels of fumonosins  (FUM) in the infected stalk, while cv. Hik-chal and Miheung-chal had the highest level of FUM.

Response: fixed (Page 1: lines 18).

L19  had a higher FUM concentration compared to cultivars infected with F. subglutinans in all cultivars tested.

Response: corrected accordingly (Page 1: line 19).

L36 Maize root rot and stalk rot have been

Response: fixed (Page 1: lines 36).

L41-42 Sometimes there is a  “,” between references and sometimes a “;”  => be consequent.

Response: We agree. Corrections have been made (Page 1: lines 42).

L42 toxigenic pathogens of maize

Response: fixed (Page 1: lines 42).

L44 of the two above-mentioned Fusarium species

Response: corrected accordingly (Page 2: line 53).

L50 this study was aimed to

Response: fixed (Page 2: lines 66).

L57 Briefly describe the greenhouse conditions (temperature, RH, light regime, etc)

Response: we appreciate this recommendation and we have carefully revised the greenhouse of our manuscript. We have mentioned the temperature, relative humidity, light and other conditions. The following sentences have been added (Page 2: lines 73-75).

The experiments of this study were performed under greenhouse conditions at 32 ± 3°C, 65-75% relative humidity, 16 h photoperiod at Kangwon National University, South Korea.”. Maize seedlings were watered when necessary”.

L62 obtained from plant pathology laboratory, Kangwon National University => what was the origin of the isolates, where and when were they collected. In addition, the experiments are done with one isolate from both species, is this representative, what is the variation between isolates within one species?

Response: we agree with the reviewer that it is necessary to describe the origin of the fungal species. In addition, it is important to note that our base for choosing the two Fusarium species is based on the previous study [23] that the mentioned species have been found the most dominant in the study area. Hence, the following sentence has been added in the revised manuscript (Page 2: lines 80-82).

“The above-mentioned Fusarium species were found to be the most dominant fungal species isolated from stalk rot of maize in Gangwon province, South Korea [23]”.

L68 was investigated. The agressiveness of the test  pathogens

Response: We agree. Corrections have been made (Page 2: lines 87).

L87 where 0 = 0, => strange sentence

Response: fixed (Page 3: lines 115).

L100 cultivar’s means of disease severity index

Response: We agree. Moreover, we have replacedmean disease severity index” with “mean percent severity index (PSI)” to keep consistency throughout the manuscript (Page 3: lines 128-131).

L130 4°C, the “4” is in another font?

Response: fixed (Page 4: lines 172).

L132 0.01 M => Sometimes there is a space between the number and the “M” and sometimes not => be consequent.

Response: We appreciate the reviewer comment and we have corrected the manuscript accordingly (Page 4: lines 174).

L167-L172 This section can be removed, you can immediately start with the results.

Response: Normally, this section was included for clarity. Nevertheless, to avoid redundancy, we have accepted the comment and the section have been removed in the revised manuscript(Page 4: lines 208).

L175 “and” not in Italics

Response: Thank you for this comment. We have carefully revised the manuscript (pages 4: 211).

L176 Here you write that you analyse the data with an ANOVA, while in “2.5 Statistical analysis” you do not mention this?

Response: Actually we have mentioned how we analyzed the data in section 2.6. The PROC GLM procedure of SAS (SAS software version 9.2) has been mentioned. We agree with the reviewer and we have avoided the redundancy in the result section.  (Pages: 4, 6).

L177 You say that there is no significant interaction between “cultivar” and “species” =>  So, in a post-hoc test you have to test for differences between cultivars and differences between species. You do not have to split the analysis per species as in Table 1. In case there was interaction you should analysis the severity index per Fusarium species, but in case there is no interaction you do not have to split….

Response: We highly appreciate the reviewer comment regarding the table presentation. We agree to change the interaction table and we have presented the table displaying the main effects (Table 1-3). 

L199 The results of the study

Response: Fixed (Page 4: lines 212).

L201 Remarkably, plants infected with F. temperatum had => Why is it “remarkably”? Please discuss

Response: We appreciate the reviewer comment. Generally, our data showed that there is a slightly higher mean FUM in F. temperatum infected stalks than F. Subglutinans. However, knowing such a slight difference between the two species it would have been better rather not to use the word remarkable. Hence, we have removed it in the revised manuscript (Page 6: lines 269).

L203 De Hack Chal accumulated comparatively low levels of FUM in…

Response: Fixed (Page 6: lines 271).

L205 De Hack Chal had substantial higher concentrations of FUM

Response: Fixedsubstantially higher” (Page 6: lines 273).

L206 there were positive correlation between stalk rot severity and FUM concentrations on each cultivar in both Fusarium species => OK this is true, but in the discussion you see that Fusarium stalk rot ratings can be used a measure for toxins in breeding. I think this need further discussion since it I know some examples where FUM concentrations do not correlate with stalk rot incidence…

Response: We appreciate the reviewer recommendation and the following sentences have been added (Page 9: lines 451-468).

“Similarly, a highly significant correlation between visual rot symptom and FUM content has been previously reported [44, 45].  Furthermore, Butrón et al. [46] reported the Fum contamination of the tested genotypes was similar across environments. Hence, the present study showing the significant correlation between stalk rot ratings and FUM levels signifies Fusarium stalk rot ratings can be an alternative tool to indirectly select maize cultivars or breeding lines having low FUM. In contrast, the previous report [47] noted that there is an interaction effect of cultivars and environment for resistance to both rot and fumonisin contamination indicating FUM production is environment dependent [46, 47]”.

L207 for both Fusarium

Response: Fixed (Page 5: lines 244).

L215 Figure 2: The symbol for correlation is “R” or “r” but not R² + mention the unit of FUM concentration in the figure + use the same x-axis in both figures.

Response: We appreciate the reviewer recommendation. However, the reviewer 1 has concern about redundancy that the graph talks about the two tables, table 1 and table 2, we have deleted the graph based on the reviewer 1 request. Furthermore, we have added the r value  in the paragraph following the reviewer1  request (Page 6: lines 276-277).

“cultivar for F. subglutinans (r=0.8269) and F. temperatum (r=0.8243)”

L215 apparently the levels of FUM are never higher than 2000 => do you have an explanation for this?

Response: The level of Fum presented in the manuscript is a mean value fumonisins (sum of B1, B2, B3). Obviously, the level of FUM in the present study by the study Fusarium species can be taken as a high FUM value compared to the previous FUM reports by the same species. For instance, Wang and his coworkers (2013) reported 166.89 μg/g FUM in maize grain sample, with massively more FB2 mycotoxin (2.8 to 108.8fold) than FB1. However, it is important to note that mycotoxin contamination like FUM is dependent on environment, and isolated hosts.

L222 resulted in typical discolored symptoms on primary and secondary roots of the tested cultivars

Response: Fixed (Page 6: lines 288-89).

L225 difference between the tested

Response: Fixed (Page 6: lines 292).

L227 indicated by the lower infection level..   To the contrary, In contrast,

Response: Fixed (Page 6: lines 294).

L232 Table 3, same remark as above, if there is no interaction you do not have to analyse the varieties for each species separately.

Response: Fixed (Page 7: lines 362).

L244 had a higher PPO activity

Response: Fixed (Page 7: lines 373).

L256 TPC was not enhanced reduced => very strange sentence.

Response: Fixed (Page 8: lines 395).

L280 resistant maize genotypes as it caused the required

Response: Fixed (Page 9: lines 425).

L283 F. subglutinans and F. temperatum (and not in Italics) (also on line 292, 315)

Response: Fixed (lines: 428, 437, 476 ).

L293 Among the tested cultivars, cultivar De Hack Chal accumulated comparatively low levels of FUM in the infected stalks while cultivar Hikchal and Miheung-chal had a substantial higher concentration of FUM in  the infected stalks, regardless of the Fusarium species

Response: Fixed (Page 9: lines 438-39).

L296 was almost close to

Response: Fixed (Page 9: lines 441).

L299 measures should

Response: Fixed (Page 9: lines 444).

L305 produced mycotoxins

Response: Fixed (Page 9: lines 450).

L311 to be more aggressive

Response: Fixed (Page 10: lines 472).

L316 while cultivars

Response: Fixed (Page 10: lines 477).

L318 to have many advantages => you say many advantages, can you give some of them, in my opinion you just order the maize varieties  which is a very common and simplistic method?

Response: one of the advantages, as far as we know, is there is no requirement of prior knowledge on the genetic information of a cultivar to be tested. Off course, we share the reviewer opinion regarding the simplicity.

L319 prior knowledge on the genetic

Response: Fixed (Page 110: lines 480).

L334 (43, 49]

Response: Fixed (Page 110: lines 495).

Reviewer 2 Report

General comments

There are still a lot of typos in the manuscript, below I listed some of them. So, please read the manuscript carefully to avoid typos.

In addition, you performed the experiments with one isolate from both species. I would be better to include more isolates, to see the differences between isolates.

Your FUM levels are never higher than 2000 µg/g, so even in case the disease pressure is very high the values are never above the limit, so is there a problem?

What are the average infection levels expected under field conditions?

Specific comments

L12 with fumonosins production

L13 to artificially inoculated F. subglutinans and

L14 maize cultivars tested

L15 stalk rot severity, while

L17 low levels of fumonosins  (FUM) in the infected stalk, while cv. Hik-chal and Miheung-chal had the highest level of FUM.

L19  had a higher FUM concentration compared to cultivars infected with F. subglutinans in all cultivars tested.

L36 Maize root rot and stalk rot have been

L41-42 Sometimes there is a  “,” between references and sometimes a “;”  => be consequent.

L42 toxigenic pathogens of maize

L44 of the two above-mentioned Fusarium species

L50 this study was aimed to

L57 Briefly describe the greenhouse conditions (temperature, RH, light regime, etc)

L62 obtained from plant pathology laboratory, Kangwon National University => what was the origin of the isolates, where and when were they collected. In addition, the experiments are done with one isolate from both species, is this representative, what is the variation between isolates within one species?

L68 was investigated. The agressiveness of the test  pathogens

L87 where 0 = 0, => strange sentence

L100 cultivar’s means of disease severity index

L130 4°C, the “4” is in another font?

L132 0.01 M => Sometimes there is a space between the number and the “M” and sometimes not => be consequent.

L167-L172 This section can be removed, you can immediately start with the results.

L175 “and” not in Italics

L176 Here you write that you analyse the data with an ANOVA, while in “2.5 Statistical analysis” you do not mention this?

L177 You say that there is no significant interaction between “cultivar” and “species” =>  So, in a post-hoc test you have to test for differences between cultivars and differences between species. You do not have to split the analysis per species as in Table 1. In case there was interaction you should analysis the severity index per Fusarium species, but in case there is no interaction you do not have to split….

L199 The results of the study

L201 Remarkably, plants infected with F. temperatum had => Why is it “remarkably”? Please discuss

L203 De Hack Chal accumulated comparatively low levels of FUM in…

L205 De Hack Chal had substantial higher concentrations of FUM

L206 there were positive correlation between stalk rot severity and FUM concentrations on each cultivar in both Fusarium species => OK this is true, but in the discussion you see that Fusarium stalk rot ratings can be used a measure for toxins in breeding. I think this need further discussion since it I know some examples where FUM concentrations do not correlate with stalk rot incidence…

L207 for both Fusarium

L215 Figure 2: The symbol for correlation is “R” or “r” but not R² + mention the unit of FUM concentration in the figure + use the same x-axis in both figures.

L215 apparently the levels of FUM are never higher than 2000 => do you have an explanation for this?

L222 resulted in typical discolored symptoms on primary and secondary roots of the tested cultivars L225 difference between the tested

L227 indicated by the lower infection level..   To the contrary, In contrast,

L232 Table 3, same remark as above, if there is no interaction you do not have to analyse the varieties for each species separately.

L244 had a higher PPO activity

L256 TPC was not enhanced reduced => very strange sentence.

L280 resistant maize genotypes as it caused the required

L283 F. subglutinans and F. temperatum (and not in Italics) (also on line 292, 315)

L293 Among the tested cultivars, cultivar De Hack Chal accumulated comparatively low levels of FUM in the infected stalks while cultivar Hikchal and Miheung-chal had a substantial higher concentration of FUM in  the infected stalks, regardless of the Fusarium species

L296 was almost close to

L299 measures should

L305 produced mycotoxins

L311 to be more aggressive

L316 while cultivars

L318 to have many advantages => you say many advantages, can you give some of them, in my opinion you just order the maize varieties  which is a very common and simplistic method?

L319 prior knowledge on the genetic

L334 (43, 49]

Author Response

Response to reviewers

Dear Editor and Reviewers,

Thank you very much for your quick response and valuable comments provided. We agree with all comments and we have revised the manuscript accordingly. Please find below a point-by-point response to the reviewers’ comments (in red).

We look forward to hearing from you soon.

Sincerely,

Youn Su Lee, Ph.D., Professor

College of Agriculture and Life Sciences

Kangwon National University, Chuncheon 24341

Republic of Korea (South Korea)

Tel: (+82)-33-250-6417, CP: (+82)-10-4131-6417

[email protected][email protected]

Vice-Chairman, Asian Mycological Association

Korean Representative, International Mycological Association

Past President, Korean Society of Mycology

Past Vice-President, Korean Society of Plant Pathology

We appreciate the reviewer’s comments and the followings are our point-by-point responses

Response to Reviewer 1 Comments

Point 1: English language and style are fine/minor spell check required

Response 1: we have carefully revised the manuscript. We hope we have improved it.

Point 2: I have two concerns about the paper:

The introduction must explain why PPO, POD and phenolic content have been selected as markers to distinghish among suscebtible and resistant genotypes in control and infected plants. It is well known that after Fusarium infection several metabolites are accumulated (see for example the recent review Frontiers in Plant Sciences 8:1774. doi: 10.3389/fpls.2017.01774)

The correlation are not presented in the correct form. I suggest to delete Fig. 2 because presents data already reported in Tables 1 and 2 and to add the R2 values at line 207.

Response 2: we agree with the reviewer that it is important to have an explanation in the introduction about the necessity of PPO, POD and phenolic content in plant-pathogen interactions. Hence, the following sentence has been added in the revised manuscript (Page 2: lines 58-65).

PPO, POD and TPC are among the most commonly reported defense responses during the interaction between a host plant and a pathogen and they have been reported as a marker for resistance to pathogen infection [16-18]. PPO is a nuclear encoded enzyme that oxidation of phenolic substances to form more toxic substances and some of them function in cross-linking reactions [17, 19]. The role of peroxidases activity has been reported in an array of defense-related processes, mainly through direct toxic effect and lignifications [20, 21]. Phenolic compounds have direct antimicrobial activity and form barriers through cell wall lignifications to prevent pathogen encroachment [18, 22]. ”.

With regard to the correlation graph, we agree that the graph normally talks about the two tables, table 1 and table 2. However, we believe that the graph enable the readers to easily catch-up the relationship about the severity of stalk rot and fumonisin accumulation in the tested Korean maize cultivars. Thus, to satisfy the reviewer concern, we have deleted the graph. Furthermore, we have added the r value (based on reviewer 2 request R2 changed to r) in the paragraph following the reviewer request (Page 6: lines 276-77).

“cultivar for F. subglutinans (r=0.8269) and F. temperatum (r=0.8243)”

Response to Reviewer 2 Comments

Point 1: Moderate English changes required

Response 1: we have carefully revised the manuscript and corrected the grammatical mistakes. We hope we have improved it.

General comments

Point 2: There are still a lot of typos in the manuscript, below I listed some of them. So, please read the manuscript carefully to avoid typos.

Response 2: we really appreciate the reviewer for detail review and we have carefully revised the manuscript accordingly.

Point 3: In addition, you performed the experiments with one isolate from both species. I would be better to include more isolates, to see the differences between isolates.

Response 3: We would like the reviewer know that our base for choosing the two Fusarium species is based on the previous study [23] that the mentioned species have been found the most dominant in the study area. Hence, to avoid confusion, the following sentence has been added in the revised manuscript (Page 2: lines 80-82).

“The above-mentioned Fusarium species were found to be the most dominant fungal species isolated from stalk rot of maize in Gangwon province, South Korea [23]”.

Similar to this question, the reviewer has asked the the origin of the isolates, where and when were they collected as it was indicated in the specific comments L62.

We agree with the reviewer that it is necessary to describe the origin of the fungal species. In addition, it is important to note that our base for choosing the two Fusarium species is based on the previous study [23] that the mentioned species have been found the most dominant in the study area.

Point 4: Your FUM levels are never higher than 2000 µg/g, so even in case the disease pressure is very high the values are never above the limit, so is there a problem? What are the average infection levels expected under field conditions?

Response 4: The level of Fum presented in our study is a mean value fumonisins (sum of B1, B2, B3). Obviously, the level of FUM in the present study by the study Fusarium species can be taken as a high FUM value compared to the previous FUM reports by the same species. For instance, Wang and his co-workers (2013) reported 166.89 μg/g FUM in maize grain sample, with massively more FB2 mycotoxin (2.8 to 108.8fold) than FB1. However, it is important to note that mycotoxin contamination like FUM is dependent on environment, and isolated hosts.

Wang and his co-workers (2013): https://onlinelibrary.wiley.com/doi/pdf/10.1111/jph.12164

Specific comments

 L12 with fumonosins production

Response: corrections have been made (Page 1: line 12). Morover, the word fumonosin was spelt wrongly in the original manuscript but now it has been spelt correctly as fumonisin in the revised manuscript.

L13 to artificially inoculated F. subglutinans and

Response: corrections have been made (Page 1: line 13).

L14 maize cultivars tested

Response: corrections have been made (Page 1: line 14).

L15 stalk rot severity, while

Response: fixed (Page 1: lines 17).

L17 low levels of fumonosins  (FUM) in the infected stalk, while cv. Hik-chal and Miheung-chal had the highest level of FUM.

Response: fixed (Page 1: lines 18).

L19  had a higher FUM concentration compared to cultivars infected with F. subglutinans in all cultivars tested.

Response: corrected accordingly (Page 1: line 19).

L36 Maize root rot and stalk rot have been

Response: fixed (Page 1: lines 36).

L41-42 Sometimes there is a  “,” between references and sometimes a “;”  => be consequent.

Response: We agree. Corrections have been made (Page 1: lines 42).

L42 toxigenic pathogens of maize

Response: fixed (Page 1: lines 42).

L44 of the two above-mentioned Fusarium species

Response: corrected accordingly (Page 2: line 53).

L50 this study was aimed to

Response: fixed (Page 2: lines 66).

L57 Briefly describe the greenhouse conditions (temperature, RH, light regime, etc)

Response: we appreciate this recommendation and we have carefully revised the greenhouse of our manuscript. We have mentioned the temperature, relative humidity, light and other conditions. The following sentences have been added (Page 2: lines 73-75).

The experiments of this study were performed under greenhouse conditions at 32 ± 3°C, 65-75% relative humidity, 16 h photoperiod at Kangwon National University, South Korea.”. Maize seedlings were watered when necessary”.

L62 obtained from plant pathology laboratory, Kangwon National University => what was the origin of the isolates, where and when were they collected. In addition, the experiments are done with one isolate from both species, is this representative, what is the variation between isolates within one species?

Response: we agree with the reviewer that it is necessary to describe the origin of the fungal species. In addition, it is important to note that our base for choosing the two Fusarium species is based on the previous study [23] that the mentioned species have been found the most dominant in the study area. Hence, the following sentence has been added in the revised manuscript (Page 2: lines 80-82).

“The above-mentioned Fusarium species were found to be the most dominant fungal species isolated from stalk rot of maize in Gangwon province, South Korea [23]”.

L68 was investigated. The agressiveness of the test  pathogens

Response: We agree. Corrections have been made (Page 2: lines 87).

L87 where 0 = 0, => strange sentence

Response: fixed (Page 3: lines 115).

L100 cultivar’s means of disease severity index

Response: We agree. Moreover, we have replacedmean disease severity index” with “mean percent severity index (PSI)” to keep consistency throughout the manuscript (Page 3: lines 128-131).

L130 4°C, the “4” is in another font?

Response: fixed (Page 4: lines 172).

L132 0.01 M => Sometimes there is a space between the number and the “M” and sometimes not => be consequent.

Response: We appreciate the reviewer comment and we have corrected the manuscript accordingly (Page 4: lines 174).

L167-L172 This section can be removed, you can immediately start with the results.

Response: Normally, this section was included for clarity. Nevertheless, to avoid redundancy, we have accepted the comment and the section have been removed in the revised manuscript(Page 4: lines 208).

L175 “and” not in Italics

Response: Thank you for this comment. We have carefully revised the manuscript (pages 4: 211).

L176 Here you write that you analyse the data with an ANOVA, while in “2.5 Statistical analysis” you do not mention this?

Response: Actually we have mentioned how we analyzed the data in section 2.6. The PROC GLM procedure of SAS (SAS software version 9.2) has been mentioned. We agree with the reviewer and we have avoided the redundancy in the result section.  (Pages: 4, 6).

L177 You say that there is no significant interaction between “cultivar” and “species” =>  So, in a post-hoc test you have to test for differences between cultivars and differences between species. You do not have to split the analysis per species as in Table 1. In case there was interaction you should analysis the severity index per Fusarium species, but in case there is no interaction you do not have to split….

Response: We highly appreciate the reviewer comment regarding the table presentation. We agree to change the interaction table and we have presented the table displaying the main effects (Table 1-3). 

L199 The results of the study

Response: Fixed (Page 4: lines 212).

L201 Remarkably, plants infected with F. temperatum had => Why is it “remarkably”? Please discuss

Response: We appreciate the reviewer comment. Generally, our data showed that there is a slightly higher mean FUM in F. temperatum infected stalks than F. Subglutinans. However, knowing such a slight difference between the two species it would have been better rather not to use the word remarkable. Hence, we have removed it in the revised manuscript (Page 6: lines 269).

L203 De Hack Chal accumulated comparatively low levels of FUM in…

Response: Fixed (Page 6: lines 271).

L205 De Hack Chal had substantial higher concentrations of FUM

Response: Fixedsubstantially higher” (Page 6: lines 273).

L206 there were positive correlation between stalk rot severity and FUM concentrations on each cultivar in both Fusarium species => OK this is true, but in the discussion you see that Fusarium stalk rot ratings can be used a measure for toxins in breeding. I think this need further discussion since it I know some examples where FUM concentrations do not correlate with stalk rot incidence…

Response: We appreciate the reviewer recommendation and the following sentences have been added (Page 9: lines 451-468).

“Similarly, a highly significant correlation between visual rot symptom and FUM content has been previously reported [44, 45].  Furthermore, Butrón et al. [46] reported the Fum contamination of the tested genotypes was similar across environments. Hence, the present study showing the significant correlation between stalk rot ratings and FUM levels signifies Fusarium stalk rot ratings can be an alternative tool to indirectly select maize cultivars or breeding lines having low FUM. In contrast, the previous report [47] noted that there is an interaction effect of cultivars and environment for resistance to both rot and fumonisin contamination indicating FUM production is environment dependent [46, 47]”.

L207 for both Fusarium

Response: Fixed (Page 5: lines 244).

L215 Figure 2: The symbol for correlation is “R” or “r” but not R² + mention the unit of FUM concentration in the figure + use the same x-axis in both figures.

Response: We appreciate the reviewer recommendation. However, the reviewer 1 has concern about redundancy that the graph talks about the two tables, table 1 and table 2, we have deleted the graph based on the reviewer 1 request. Furthermore, we have added the r value  in the paragraph following the reviewer1  request (Page 6: lines 276-277).

“cultivar for F. subglutinans (r=0.8269) and F. temperatum (r=0.8243)

L215 apparently the levels of FUM are never higher than 2000 => do you have an explanation for this?

Response: The level of Fum presented in the manuscript is a mean value fumonisins (sum of B1, B2, B3). Obviously, the level of FUM in the present study by the study Fusarium species can be taken as a high FUM value compared to the previous FUM reports by the same species. For instance, Wang and his coworkers (2013) reported 166.89 μg/g FUM in maize grain sample, with massively more FB2 mycotoxin (2.8 to 108.8fold) than FB1. However, it is important to note that mycotoxin contamination like FUM is dependent on environment, and isolated hosts.

L222 resulted in typical discolored symptoms on primary and secondary roots of the tested cultivars

Response: Fixed (Page 6: lines 288-89).

L225 difference between the tested

Response: Fixed (Page 6: lines 292).

L227 indicated by the lower infection level..   To the contrary, In contrast,

Response: Fixed (Page 6: lines 294).

L232 Table 3, same remark as above, if there is no interaction you do not have to analyse the varieties for each species separately.

Response: Fixed (Page 7: lines 362).

L244 had a higher PPO activity

Response: Fixed (Page 7: lines 373).

L256 TPC was not enhanced reduced => very strange sentence.

Response: Fixed (Page 8: lines 395).

L280 resistant maize genotypes as it caused the required

Response: Fixed (Page 9: lines 425).

L283 F. subglutinans and F. temperatum (and not in Italics) (also on line 292, 315)

Response: Fixed (lines: 428, 437, 476 ).

L293 Among the tested cultivars, cultivar De Hack Chal accumulated comparatively low levels of FUM in the infected stalks while cultivar Hikchal and Miheung-chal had a substantial higher concentration of FUM in  the infected stalks, regardless of the Fusarium species

Response: Fixed (Page 9: lines 438-39).

L296 was almost close to

Response: Fixed (Page 9: lines 441).

L299 measures should

Response: Fixed (Page 9: lines 444).

L305 produced mycotoxins

Response: Fixed (Page 9: lines 450).

L311 to be more aggressive

Response: Fixed (Page 10: lines 472).

L316 while cultivars

Response: Fixed (Page 10: lines 477).

L318 to have many advantages => you say many advantages, can you give some of them, in my opinion you just order the maize varieties  which is a very common and simplistic method?

Response: one of the advantages, as far as we know, is there is no requirement of prior knowledge on the genetic information of a cultivar to be tested. Off course, we share the reviewer opinion regarding the simplicity.

L319 prior knowledge on the genetic

Response: Fixed (Page 110: lines 480).

L334 (43, 49]

Response: Fixed (Page 110: lines 495).
